Obama chez moi! The invasion of metropolitan France by the land planarian Obama nungara (Platyhelminthes, Geoplanidae)

http://orcid.org/0000-0002-7155-4540 Justine Jean-Lou 1 justine@mnhn.fr
Winsor Leigh 2
Gey Delphine 3
Gros Pierre 4
http://orcid.org/0000-0002-8077-4879 Thévenot Jessica 5
1 ISYEB, Institut de Systématique Évolution Biodiversité, UMR7205 CNRS, EPHE, MNHN, UPMC, Université des Antilles, Muséum National d’Histoire Naturelle , Paris , France
2 College of Science and Engineering, James Cook University , Townsville, QLD , Australia
3 Service de Systématique Moléculaire, Museum National d’Histoire Naturelle , Paris , France
4 Amateur Naturalist , Cagnes-sur-Mer , France
5 UMS Patrinat (CNRS–AFB–MNHN), Muséum National d’Histoire Naturelle , Paris , France
Tatarinova Tatiana
Electronic publication date: 2020 Feb 6
Publication date: 2020
Volume: 8
Electronic Location ID: e8385
Received 2019 Aug 8; Accepted 2019 Dec 11
Copyright: © 2020 Justine et al.
Copyright year: 2020
Copyright holder: Justine et al.
License: This is an open access article distributed under the terms of the Creative Commons Attribution License, which permits unrestricted use, distribution, reproduction and adaptation in any medium and for any purpose provided that it is properly attributed. For attribution, the original author(s), title, publication source (PeerJ) and either DOI or URL of the article must be cited.
License URL: https://creativecommons.org/licenses/by/4.0/

Keywords: Platyhelminthes, France, Alien invasive species, Land planarians, Barcoding, Citizen science

Funding: Actions Thématiques du Muséum (ATM) from Muséum National d’Histoire Naturelle, Paris, France This work was supported by Actions Thématiques du Muséum (ATM) from Muséum National d’Histoire Naturelle, Paris, France. The funders had no role in study design, data collection and analysis, decision to publish, or preparation of the manuscript.

==============================
Background

Obama nungara is a species of land flatworm originating from South America; the species was recently described and distinguished from a similar species, Obama marmorata. Obama nungara has invaded several countries of Europe, but the extent of the invasion has not been thoroughly mapped.

Methods

In this article, based on a five and a half-year survey undertaken by citizen science, which yielded 530 records from 2013 to 2018, we analysed information about the invasion of Metropolitan France by O. nungara. We also investigated the variability of newly obtained cytochrome c oxidase 1 (COI) sequences of specimens from France, Italy and Switzerland.

Results

Obama nungara was recorded from 72 of the 96 Departments of Metropolitan France. The species is especially abundant along the Atlantic coast, from the Spanish border to Brittany, and along the Mediterranean coast, from the Spanish border to the Italian border. More than half of the records were from an altitude below 50 m, and no record was from above 500 m; mountainous regions such as the Alps, Pyrenees and Massif Central are not invaded. Local abundance can be impressive, with 100 of specimens found in a small garden. An analysis of our new COI sequences, combined with published sequences of specimens from several countries, confirmed that three clades comprise the species. The first clade, ‘Brazil’, is currently confined to this country in South America; the second clade, ‘Argentina 2’, was found in Argentina and in Europe, only in Spain; and the third, ‘Argentina 1’, was found in Argentina and in Europe, in Spain, Portugal, France, UK, Italy, Belgium, and Switzerland. This suggests that two clades of O. nungara from Argentina have invaded Europe, with one widely spread.

Discussion

The present findings strongly suggest that O. nungara is a highly invasive species and that the population which has invaded several countries in Europe comes from Argentina. The wide dispersion of the species and its reported local abundance, combined with the predatory character of the species, make O. nungara a potential threat to the biodiversity and ecology of the native soil fauna in Europe, and probably the most threatening species of all invasive land planarians present in Europe.

Introduction

Several land planarians have been reported as invasive alien species in Europe (Table 1). Each of these species shows distinctive features. Some are remarkable by their considerable size, such as Bipalium kewense and Diversibipalium multilineatum, which can reach 30 cm in length (Justine et al., 2018b). Some have invaded only parts of Europe, such as Arthurdendyus triangulatus, mostly confined to the northern part of the British Isles (Jones et al., 2001). Some are relatively rare, such as Caenoplana coerulea (Justine, Thévenot & Winsor, 2014), or inconspicuous, such as the small species Marionfyfea adventor (Jones & Sluys, 2016). Only one species, A. triangulatus, has been added, very recently, to the list of invasive alien species of Union concern (European Union, 2019).

Table 1 Invasive land planarians found in Europe, authors of taxa and key references.

This table provides complete information about authors and taxa and combination, thus making the general text lighter. Sluys (2016) listed additional species with limited records and information: Artioposthia exulans Dendy (1901), Australoplana sanguinea (Moseley, 1877), Dolichoplana striata Moseley (1877), and Kontikia andersoni Jones (1981). This table updates a similar table published in Justine et al. (2018b). Synonyms are limited to binomials, various ‘sp.’ in GenBank not listed.

Taxon and authors	Synonyms	References for taxon	Main references for presence in Europe	
Obama nungara Carbayo, Álvarez-Presas, Jones & Riutort, 2016	Obama marmorata pro parte.	Carbayo et al. (2016)	Carbayo et al. (2016); Lago-Barcia et al. (2019); this article	
Arthurdendyus triangulatus (Dendy, 1896) Jones, 1999	Artioposthia triangulata	Dendy (1895); Jones (1999)	Boag et al. (1994)	
Australopacifica atrata (Steel, 1897)	Geoplana atrata; Parakontikia atrata; Kontikia atrata	Steel (1897)	Jones (2019)	
Bipalium kewense Moseley, 1878		Moseley (1878)	Justine et al. (2018b)	
Caenoplana bicolor (Graff, 1899) Winsor, 1991	Geoplana bicolor	Von Graff (1899); Winsor (1991)	Álvarez-Presas et al. (2014)	
Caenoplana coerulea Moseley, 1877		Moseley (1877)	Álvarez-Presas et al. (2014); Breugelmans et al. (2012)	
Diversibipalium ‘black’		Justine et al. (2018b)	Justine et al. (2018b)	
Diversibipalium multilineatum (Makino & Shirasawa, 1983) Kubota & Kawakatsu, 2010	Bipalium multilineatum	Makino & Shirasawa (1983); Kubota & Kawakatsu (2010)	Justine et al. (2018b); Mazza et al. (2016)	
Marionfyfea adventor Jones & Sluys, 2016		Jones & Sluys (2016)	Jones & Sluys (2016)	
Parakontikia ventrolineata (Dendy, 1892) Winsor, 1991	Kontikia ventrolineata	Dendy (1892); Winsor (1991)	Álvarez-Presas et al. (2014)	
Platydemus manokwari De Beauchamp, 1963		De Beauchamp (1962)	Justine et al. (2014)	

This article concentrates on one species, Obama nungara, which has already been recorded in several European countries (Carbayo et al., 2016). So far, there are published records from Guernsey Island, UK, Spain, Portugal, France, Belgium and Italy, valid unpublished records from Ireland to Madeira Island, and an alleged record from the Netherlands (Table 2). In this article, we add a record from Switzerland. No record is known from Germany (Sluys, 2019), or any country East of Germany.

Table 2 Records of Obama nungara in Europe.

The year is indicated as the date of the first record or first specimen mentioned in the reference; actual presence of the species may predate the year mentioned.

Country or territory	Year	References	
Guernsey Island	2008	Carbayo et al. (2016)	
Mainland United Kingdom	2009	Carbayo et al. (2016)	
Spain	2010	Carbayo et al. (2016)	
Portugal	N/A	Lago-Barcia et al. (2019)	
Metropolitan France (excluding Corsica Island)	2013	Justine, Thévenot & Winsor (2014); this article	
Corsica	2013	This article	
Italy	2012	Carbayo et al. (2016); this article	
Switzerland	2014	This article	
Ireland	2009	H. Jones, 2019, personal communication; collection and identification: R. Anderson.	
Madeira Island	2018	H. Jones, 2019, personal communication	
The Netherlands
In the UK, in a pot plant allegedly ‘imported from the Netherlands’	2016	Aldred (2016)	
Belgium	2017	Soors et al. (2019)	

Research on invasive species calls for the participation of the public in order to obtain information on a detailed scale. Like our previous studies on invasive land flatworms (Justine, Thévenot & Winsor, 2014; Justine et al., 2014, 2015, 2018b), this research on O. nungara was based on hundreds of reports from a large number of citizens. We acknowledge the immense value of these records from citizen science (Kullenberg & Kasperowski, 2016); however, in this article, we point out some limitations concerning the accuracy of citizen science records due to bias associated with media buzz.

The identification of land planarians is difficult, and the name O. nungara itself was assigned to the species as recently as 2016 (Carbayo et al., 2016). In this article, in addition to the information based on data from citizen science, we sequenced a number of specimens from various regions of France to obtain cytochrome c oxidase 1 (COI) barcodes, and analysed our sequences in the light of recent research on the geographic origin of this invasive species (Lago-Barcia et al., 2019).

The results of this analysis suggest that the specimens found in France, Italy and Switzerland, and sequenced by us, have their geographical origin in Argentina.

Materials and Methods

Citizen science and collection of information

Data were collected from mid-2013 to the end of 2018, a period of 5.5 years. For this study, we used the same method as for our previous research on invasive flatworms (Justine, Thévenot & Winsor, 2014; Justine et al., 2014, 2015, 2018b). Briefly, a blog (Justine, 2019b) and a Twitter account (https://twitter.com/Plathelminthe4) were instrumental in collecting and transmitting information. Reports of sightings of land planarians were received from the general public and sometimes from professionals, generally by email and, since 2018, from a dedicated web page (http://eee.mnhn.fr/). Inaccurate records (slugs, earthworms, leeches and others) were eliminated. Two waves of reports were received in 2014 and 2015 when we published our articles on Platydemus manokwari (Justine et al., 2014, 2015) and when the media reported the presence of this highly invasive species in metropolitan France. Many reports were labelled as ‘Platydemus manokwari’ or ‘New Guinea Flatworm’ by the public; none were P. manokwari in the wild in metropolitan France, and in most cases were in fact O. nungara.

Photographs received from non-professionals were studied. O. nungara is typically 5–8 cm in length, with a brownish dorsal side (varying among specimens from orange to almost black), a beige sole, and a body shape with a pointed head. In metropolitan France, confusion on the basis of photographs sent by amateurs is possible only with (1) Parakontikia ventrolineata, but this species is smaller and has typical longitudinal lines on its back and belly; these lines are often not easily visible on photographs taken by amateurs but can be seen after enhancing the contrast, and (2) P. manokwari, but this species has a typical body shape, two conspicuous eyes, and a clear, pale single longitudinal line on its back. Conversely, O. nungara cannot possibly be confused with Caenoplana bicolor (typical bright yellow-orange dorsal side with two black longitudinal lines), C. coerulea (typical well-visible white line on the back, blue sole), and the bipaliines (typical head shape, and large size for B. kewense and D. multilineatum). Confusion of O. nungara with smaller species, such as M. adventor, Rhynchodemus spp. or Microplana spp. is unlikely. Finally, we are confident that all records mentioned in this article are O. nungara.

A few non-professional contributors decided, on their own initiative, to do much more than what was requested and registered the number of worms they were killing over certain periods of time. Some examples of this information are included here. While we are aware that these observations were not performed by scientists, we see no reason why they may be erroneous, and thus we believe that they are be a valuable contribution to our scientific findings.

In our previous article on bipaliines (Justine et al., 2018b), after seeking authorisation from non-professionals, we made a considerable effort to publish all available photographs. This was time-consuming, since formal authorisation had to be collected sometimes up to 5 years after an email was received. In this article, we received a far higher number of citizen science reports than in our previous article on bipaliines (530 records vs. 117). We decided that publishing all the photographic information and data would make the Supplemental Files too large and was probably not beneficial. Instead, we selected the best and most informative photographs.

Our aim was originally to collect information for metropolitan France, but a few records were also obtained from neighbouring countries, namely Switzerland and Italy.

In an open science perspective, we have made all 530 records obtained in France public, as Supplemental File 1; records have been anonymised, that is the names of private persons have been deleted. Data will also be transferred to national databases on biodiversity and then to the Global Biodiversity Information Facility.

Collection of specimens

In some cases, after examining a photograph, we solicited specimens from the non-professionals or professionals who reported sightings. Specimens were sent either alive or in ethanol, registered in the collections of the Muséum National d’Histoire Naturelle, Paris (MNHN), and processed for molecular analysis.

When specimens were obtained alive, they were fixed in near boiling water and preserved in 95% ethanol.

Molecular sequences

For molecular analysis, a small piece of the body (1–3 mm3) was taken from the lateral edge of ethanol-fixed individuals. Genomic DNA was extracted using the QIAamp DNA Mini Kit (Qiagen, Hilden, Germany). Two sets of primers were used to amplify the COI gene. A fragment of 424 bp (designated in this text as ‘short sequence’) was amplified with the primers JB3 (=COI-ASmit1) (forward 5′-TTTTTTGGGCATCCTGAGGTTTAT-3′) and JB4.5 (=COI-ASmit2) (reverse 5′-TAAAGAAAGAACATAATGAAAATG-3′) (Bowles, Blair & McManus, 1995; Littlewood, Rohde & Clough, 1997). PCR reactions were performed in 20 μl, containing 1 ng of DNA, 1× CoralLoad PCR buffer, three mM MgCl2, 66 µm of each dNTP, 0.15 μm of each primer and 0.5 units of Taq DNA polymerase (Qiagen, Hilden, Germany). The amplification protocol was: 4′ at 94 °C, followed by 40 cycles of 94 °C for 30″, 48 °C for 40″, 72 °C for 50″, with a final extension at 72 °C for 7′. A fragment of 825 bp was amplified with the primers BarS (forward 5′-GTTATGCCTGTAATGATTG-3′) (Álvarez-Presas et al., 2011) and COIR (reverse 5′-CCWGTYARMCCHCCWAYAGTAAA-3′) (Lázaro et al., 2009; Mateos et al., 2013). PCR products were purified and sequenced in both directions on a 96-capillary 3730xl DNA Analyzer sequencer (Applied Biosystems, Foster City, CA, USA). Results of both analyses were concatenated to obtain a COI sequence of 909 bp in length (designated in this text as ‘long sequence’). Sequences were edited using CodonCode Aligner software (CodonCode Corporation, Dedham, MA, USA), compared to the GenBank database content using BLAST, and deposited in GenBank under Accession Number MN529561–MN529582. For several specimens, only ‘short’ sequences were obtained (Table 3).

Table 3 Specimens of O. nungara used for molecular analysis.

All specimens were adults, except JL067, received as cocoons. All records are from France, except one record from Italy and one from Switzerland.

GenBank	MNHN	Date	Locality	Department or Country	COI	Collector	
MN529561	JL055	30/04/2013	Cagnes-sur-Mer	Alpes-Maritimes	Long	Pierre Gros	
MN529562	JL057B	17/05/2013	Lamballe	Côtes-d’Armor	Long	Benoît L’Hotellier	
MN529563	JL067	10/10/2013	Guiclan	Finistère	Long	Madame Stephan (Cocoon)	
MN529564	JL080	19/08/2013	Salernes	Var	Long	Daniel Juif	
MN529565	JL092A	26/11/2013	Montauban	Tarn-et-Garonne	Long	Céline Tan	
MN529566	JL094	04/12/2013	Paris	Paris	Long	Xavier Japiot (VILLE DE PARIS)	
MN529567	JL098A	27/03/2014	Les Matelles	Hérault	Short	Benoit Jaillard (INRA)	
MN529568	JL098C	27/03/2014	Les Matelles	Hérault	Short	Benoit Jaillard (INRA)	
MN529569	JL101	02/04/2014	Hérouville-Saint-Clair	Calvados	Short	Daniel Loisel (FREDON)	
MN529570	JL103	21/06/2014	Ile de Ré	Charente-Maritime	Short	Félix Bécheau	
MN529571	JL106	02/04/2014	Metz	Moselle	Long	Charlie Sommer (FREDON)	
MN529572	JL109A	29/03/2014	Rome	Italy	Long	Di Pompeo et al.	
MN529573	JL145	06/05/2014	La Plaine Saint Denis	Seine-Saint-Denis	Short	Dhyma Gomez	
MN529574	JL147A	16/05/2014	La Flocellière (Sèvremont)	Vendée	Short	Jocelyn Foucher	
MN529575	JL152	24/06/2014	Cagnes-sur-Mer	Alpes-Maritimes	Long	Pierre Gros	
MN529576	JL168A	14/08/2014	Geneva	Switzerland	Long	Corinne Jasquelin (DGNP, Genève)	
MN529577	JL193	19/10/2014	Lay-Saint-Christophe	Meurthe-et-Moselle	Long	Emeline Notte (AREXHOR)	
MN529578	JL196	07/11/2014	Châtenay-Malabry	Hauts-de-Seine	Long	Thibault Garnier Boudier (DEPT)	
MN529579	JL243	26/03/2015	Caen	Calvados	Long	Arnaud Pudepiece (FREDON)	
MN529580	JL245	21/03/2015	Brétigny-sur-Orge	Essonne	Long	Nicolas Puillandre (MNHN)	
MN529581	JL246A	07/04/2015	Les Matelles	Hérault	Long	Benoit Jaillard (INRA) and students	
MN529582	JL259B	26/05/2015	Lyon	Rhône	Short	Cloé Laurent (VILLE DE LYON)	

Matrix

Many sequences of the COI gene of O. nungara are available in GenBank, under various species names, due to confusion and the complex nomenclatural history of the species (Álvarez-Presas et al., 2014; Carbayo et al., 2016; Lago-Barcia et al., 2015, 2019). These sequences were obtained with various primers and thus overlap only partially. We constructed a matrix following three steps. Step 1: we constructed a large preliminary matrix including almost all the sequences of O. nungara (under different names) from GenBank and selected sequences of other species of Obama. We found that only a part of the COI sequence was shared by a significant number of them. Step 2: we trimmed the matrix to keep only the shared part. Preliminary analyses with this matrix produced trees with low support, because many of the sequences from GenBank had misreads. Step 3: we deleted all sequences containing misreads. The final matrix, used in this analysis, is 255 bases in length and includes 99 sequences, namely eight sequences of seven Obama species (O. anthropophila, O. burmeisteri, O. carinata, O. josefi, O. ladislavii, O. marmorata, O. maculipunctata) and 91 sequences of O. nungara including our 22 new sequences. The matrix includes sequences of O. nungara from Argentina, Brazil, and seven countries in Europe; it is perfectly ‘clean’ with no misreading errors or blank parts, that is it provides information on 99 taxa for all 255 positions.

In an open science perspective, we have also made this matrix public as Supplemental File 2 with this article; the matrix, in MEGA format (.meg) (Kumar, Stecher & Tamura, 2016), includes information about ‘groups’ (a MEGA feature), that is the clades found in the analysis. It is also provided in FASTA format (.fasta), which does not include this information.

Trees and distances

MEGA7 (Kumar, Stecher & Tamura, 2016) was used to evaluate distances, choose models and construct trees. The best evolutionary model for the data set in our matrix, under the Bayesian Information Criterion, was the Hasegawa–Kishino–Yano model with Gamma distribution (Hasegawa, Kishino & Yano, 1985). The evolutionary history was inferred using the Maximum Likelihood (ML) method, with 1,000 bootstrap replications. The Neighbour-Joining method (Saitou & Nei, 1987) was also inferred for comparison, with 1,000 bootstrap replications. Distances within and between the clades within O. nungara and between these clades and the outgroup were computed with various methods (Tamura-3, p-distance, Kimura-2, and Maximum composite likelihood) (Kimura, 1980; Tamura, 1992; Tamura, Nei & Kumar, 2004).

Population analysis

Haplotype networks were constructed using the same matrix as for the trees, but restricted to the species O. nungara, that is a matrix with 91 taxa, 255 bases in length. We used PopART software (Leigh & Bryant, 2015). In the nexus file, traits were the geographical origin of the species, with number of traits set to nine, and trait labels set as: Argentina, Brazil, UK (including Guernsey), Portugal, Spain, Italy, Switzerland, Belgium and Metropolitan France. The method of Templeton, Crandall and Sing was used to infer relationships among samples (Clement et al., 2002).

Similarly, following the open science perspective, we have made this matrix public as Supplemental File 3, in nexus (.nex) format.

Results

Adult specimens and/or photographs of adult specimens sent to us corresponded well with the description of the external morphology and published photographs of O. nungara, including the dark and light-coloured forms (Carbayo et al., 2016). None of our specimens were examined histologically.

Morphology of O. nungara from Metropolitan France

At rest, O. nungara is flattened and leaf-shaped (Fig. 1), and when active it is elongate with parallel margins tapering gently to the anterior tip and narrowing more abruptly to a point posteriorly.

Figure 1 An example of the pale coloured form of Obama nungara at rest.

Specimen MNHN JL055 from Cagnes-sur-Mer, Alpes-Maritimes. Photo by Pierre Gros.

In the living specimens examined in this study, the dorsal ground colour varied from uniform light honey-brown to dark brown, with darker brown striae and stippling.

In light-coloured specimens (Figs. 1 and 2), poorly defined paired median dorsal longitudinal stripes of dark-brown pigment are separated by a fine light brown median zone of pale ground colour, without striae or stippling, that extend from the anterior tip to the posterior end. The medial margins of these stripes are sharp, and the outer margins diffuse and uneven. The dark pigment in the median paired stripes is concentrated over the region of the pharynx and copulatory organs (Fig. 1) in the posterior body third. In the mid body, the darkly pigmented broad transverse bars about a third of the width of the body are separated by narrow intervals of ground colour either side of the pale medial line (Fig. 2). In some specimens, these clumps of pigmentation look like ‘tiger stripes’ and may continue anteriorly towards the tip (Fig. 3).

Figure 2 Pale-yellow form of Obama nungara, feeding on a snail.

The specimen shows transverse aggregations of pigment either side of the pale mid dorsal stripe. This specimen is feeding on a snail (Theba pisana), and the dorsal puckering results from protrusion of the pharynx into the snail. Specimen from Cagnes-sur-Mer, Alpes-Maritimes. Photo by Pierre Gros.

Figure 3 Dorsal view of Obama nungara.

This specimen shows pronounced dorsal ‘tiger stripes’. Specimen from Cagnes-sur-Mer, Alpes-Maritimes. Photo by Pierre Gros.

External to these paired medial ‘tiger stripes’, brown striae comprised of aggregations of dark pigment are oriented longitudinally along the body, the colour and pattern of which give the body a fine gneissic texture. A submarginal zone of light ground colour, devoid of striae or stippling, borders the entire body almost to the darker brown anterior tip.

The ventral surface is a pale grey-cream colour (pale beige), lightest in the midline (Fig. 4).

Figure 4 Obama nungara, specimen showing dorsal and ventral sides.

The ventral surface is pale. Specimen MNHN JL055 from Cagnes-sur-Mer, Alpes-Maritimes. Photo by Pierre Gros.

A single row of eyes contours the anterior tip, crowd slightly antero-laterally, and then spread dorsally, covering about one third of the body width (Fig. 5). The eyes, especially those behind the anterior tip and those that extend dorsally, may exhibit a zone devoid of pigment surrounding each eye, resembling small halos that are more conspicuous in darker than lighter forms.

Figure 5 Obama nungara, anterior part showing eyes, right lateral view.

The ‘halo’ effect of a clear zone around the eyes is present, though relatively inconspicuous in the pale form compared to the darker form. Specimen MNHN JL055 from Cagnes-sur-Mer, Alpes-Maritimes. Photo by Pierre Gros.

The dark brown form (Fig. 6) was the most abundant colour variety of specimens examined in this study. In dark brown specimens, the pale median longitudinal stripe may almost be obliterated by the densely dark pigmented median stripes; the anterior tip is a light brown, and the eyes, behind the anterior tip and dorsally, exhibit ‘haloing’ (Fig. 7).

Figure 6 Obama nungara, dark form feeding on an earthworm.

The everted pharynx can be clearly seen partly enveloping the head of the earthworm (unidentified species). Specimen MNHN JL092 from Montauban, Tarn-et-Garonne. Photo by Pierre Gros.

Figure 7 Obama nungara, dark form anterior part showing eyes.

Antero-lateral aspect showing ‘haloing’ around the eyes. Specimen from Antibes, Alpes-Maritimes. Photo by Pierre Gros.

Living adult specimens were medium sized, ranging in length from 52 mm to 108 mm with one individual measuring 68.0 mm long, maximum width 5.5 mm, with the mouth 41.5 mm (61%) and gonopore 53 mm (77.9%) behind the anterior tip (measured on a photograph).

A number of preserved and sequenced specimens were measured; measurements are given in mm, as length (L), distance of mouth from anterior tip (G), distance of gonopore from anterior tip (G), and percentages: MNHN JL092A, L 40, M 25 (62.5%), G 33 (82.5%); MNHN JL094, L 28, M 15 (53.6%), G 20 (71.4%); MNHN JL101A, L 40, M 25 (62.5%), G 33 (82.5%); MNHN JL101B, L 48, M 28 (58.3%), G 38 (79.2%).

Cocoons were bright red when freshly laid (Fig. 8), tanning to dark brown over 3 days (Fig. 9). Cocoons took 13–17 days to hatch at about 20 °C. Three juveniles freshly emerged from their cocoon, laid by MNHN JL092 (Fig. 10), measured 10 × 1.4 mm, 7 × 1.1 mm and 4.1 × 1.3 mm. They were light cream in colour with fine dark brown stippling, with some aggregation of the fine spots towards the mid-dorsum delineating a fine discontinuous median unpigmented dorsal stripe, with an indication of dark transverse bars. The anterior tips were dark grey, and ventral surfaces pale grey.

Figure 8 Obama nungara, adult specimen with cocoon.

The cocoon has been freshly laid and measures 4.8 mm in diameter; its colour is reddish. Photo by Pierre Gros.

Figure 9 Obama nungara, adult specimen with cocoon.

The cocoon, 82 h after being laid, measures 3.7–3.9 mm diameter; its colour is now black. Cocoon laid by specimen MNHN JL92 from Montauban, Tarn-et-Garonne. Photo by Pierre Gros.

Figure 10 Obama nungara, juveniles.

The collapsed egg cocoon and the three juveniles that it contained. Cocoon laid by Specimen MNHN JL092 from Montauban, Tarn-et-Garonne. Photo by Pierre Gros.

As previously explained, we received hundreds of photographs of adult specimens taken by non-professional contributors and cannot show all images here. However, most can be retrieved from the Twitter feed of one of the authors (@Plathelminthe4). We show here only two photographs of special interest. Figure 11 is a single specimen which shows the dark-brown pattern; this single specimen was found in Paris in December 2013, and this photo was therefore used by many media channels; the specimen was deposited in our collection as MNHN JL094, and sequenced (Table 3). Figure 12 is a photograph of a plastic box with its surface covered with adult specimens, which illustrates the abundance of the species in some gardens. Figure 13 includes a sample of nine photographs and shows the variety of images of O. nungara received from various contributors, with and without scales, sometimes in the natural habitat or, on the contrary, arranged on a coloured surface by photographers.

Figure 11 Obama nungara found in Paris in December 2013.

This single specimen was found in the Bois de Vincennes, Paris and the photograph was widely used by the French Press. Specimen MNHN JL094, barcoded. Photograph by Xavier Japiot, CC-BY-SA-3.0. https://commons.wikimedia.org/wiki/File:Planaria_Geoplanidae,_3_cm_(02).JPG.

Figure 12 Large numbers of Obama nungara found in a garden.

A box filled with adult specimens collected by hand by a non-professional (Sylvain Petiet) in a single day in May 2014, in a small garden in Cabanac-et-Villagrains (Gironde). Photo by Sylvain Petiet.

Figure 13 A sample of photographs of Obama nungara in gardens, received from non-professionals.

The photographs in (A), (E) and (H): are examples of the light brown colour; others are of the dark form. Scales in (B) and (G): centimetres and millimetres; diameter of Euro 10 cent coin in (H) and (I): 19.5 mm; other images are unscaled. All authors have agreed to publication of their photographs under a CC-BY 4.0 licence. (A) Cathy Constant-Elissagaray. (B) Nicolas Armengaud. (C) Julien Silvert. (D) Frédéric Madre. (E) Benjamin Klein. (F) Françoise Bronnec. (G) Louise Lejus. (H) Fanny Tourraille. (I) Christophe and Amauray Amiand.

Information obtained from citizen science: presence

Our database of valid records of land flatworms received from mid-2013 to December 2018 contains more than 1,000 records, after elimination of species that are not flatworms. We found that 530 valid reports, that is over half of them, concerned O. nungara in Metropolitan France. This is therefore the most often recorded species. Most reports were from non-professionals, while a few were from professionals or scientists.

Figure 14 is a map of these records in metropolitan France, with each record shown as a dot on the map. The map includes general colour-coded information on altitude. The records were most abundant along the Atlantic and Mediterranean coasts; mountainous areas (Alps, Pyrenees, Massif Central, France) were devoid of records, which were mainly in the plains. The north-eastern quarter of the country, which has a colder climate, had fewer records than the other three quarters; however, there was a concentration of records around Paris.

Figure 14 Map of records of Obama nungara in Metropolitan France in the period 2013–2018.

Each red dot is a record in the period 2013–2018. Some overlap between dots occurred. Note the concentration of records along the Atlantic and Mediterranean coasts. Mountainous parts (yellow on the map) were not invaded.

In Fig. 14, a number of records may be overlooked when dots are superposed, when several records were received from the same locality, or when they were from neighbouring localities. Figure 15 is a map of metropolitan France with records shown as numbers in each Department (Departments are the main administrative divisions in France, with a current number of 96 for Metropolitan France). Among the 96 Departments, 72 (75%) had records. The Departments with the most records were Gironde and Finistère (40–80 records), on the Atlantic Coast, and most Departments along the Atlantic Coast from the Spanish border to Brittany had more than 30 records. One noticeable case is Haute–Garonne, which is not on the coast but stands out from the neighbouring Departments with more than 40 records. Only 24 Departments had no records; they are mainly in the Massif Central and the north-eastern quarter of the country. In the discussion, we briefly comment on these results in view of population densities in various Departments.

Figure 15 Map of records of Obama nungara in Metropolitan France in the period 2013–2018, shown as records in each Department.

Records are shown as number of records in each Department. The colour shows the intensity of the invasion. Note that all Departments along the Atlantic and Mediterranean coasts are heavily affected. The numbers (31 and 33) show two Departments (31: Haute-Garonne; 33: Gironde) that are mentioned in the “Discussion”.

We used our database to investigate the influence of altitude. Figure 16 shows a chart of the number of records against increasing altitude. More than half of the records were from an altitude below 50 m; records above 250 m were rare, and no records were from above 500 m.

Figure 16 Altitude of localities in which Obama nungara was recorded in Metropolitan France.

More than half of the records were below 50 m, and there were no records above 500 m.

Information obtained from citizen science: abundance

In addition to the numerous ‘one time’ records mentioned above, we received two reports from non-professionals who counted the flatworms in their gardens over a long period.

Sylvain Petiet, from Cabanac-et-Villagrains (Gironde, south-western France), collected and destroyed specimens every day in his garden from May to August 2014; he estimated the invaded part to be about 300 m2, in a larger garden about 5,000 m2 in surface area. He collected a total of 924 specimens: from 3 May 2014 to 29 May 2014 (26 days), 180 specimens (6.9/day), and from 29 May 2014 to 22 August 2014 (85 days), 744 specimens (8.7/day). After that, he reported in 2014 that flatworms were still as numerous in the garden. Figure 13 is a photograph of a box of worms collected in May 2014 by Sylvain Petiet. Interestingly, he reported in 2019 that no flatworms were present.

Michel Hir, from Brest (Finistère, north-western France), collected and destroyed specimens every day in his garden from November 2015 to May 2016 (Table 4). His garden is 175 m2 in surface area. He mentioned that the garden was enclosed by 2 m high walls and observed that specimens were only occasionally seen on the wall but never more than 40 cm in height, so we can consider that all specimens found originated from reproduction of species from his own garden, without invasion from the neighbouring areas. He collected a total of 1,442 specimens (10.2/day). After that, he reported that flatworms were still as numerous in the garden.

Table 4 Number of specimens of Obama nungara in a garden.

The specimens were collected and destroyed every day by a non-professional in a 175 m2 garden in Metropolitan France, from November 2015 to May 2016. May 2016, only first 20 days. Data by Michel Hir.

	November	December	January	February	March	April	May (20 days)	Total	
Total	477	414	110	107	52	232	50	1,442	
Mean/day	19	17	6	6	4	11	8	10	

A simple calculation from Michel Hir’s data for the highest period he recorded (November: 19 worms/day/175 m2) provides an estimate of population increase due to reproduction of 1,085 worms/hectare/day, or 0.1 worm/m2/day. Importantly, all these counts are based on adult individuals only, which suggests that the number of hatchlings was even higher (although mortality of hatchlings was not assessed, we can assume that it is not zero).

Trees and distances

A tree (Fig. 17) was constructed from a matrix built from a selection of COI sequences of O. nungara from GenBank and our own new sequences, with a selection of sequences from other species as the outgroup. The matrix had 99 taxa with 255 positions and was perfectly clean, that is without misread or blank parts.

Figure 17 Maximum-likelihood tree of Obama nungara and close relatives.

The analyses involved 99 nucleotide sequences and there was a total of 255 positions in the final dataset. The matrix included a selection of available sequences of O. nungara and sequences of six other Obama species (noted with *), selected upon the absence of misreads. The percentage of trees in which the associated taxa clustered together is shown next to the branches (ML, Maximum Likelihood; NJ, Neighbour-Joining). Three clades are visible within O. nungara: ‘Argentina 1’, ‘Argentina 2’ and ‘Brazil’, named after the country of origin of the South American sequences. Specimens from various localities in Europe, including all specimens from France, were included in the ‘Argentina 1’ clade. A few specimens from Spain were included in the ‘Argentina 2’ clade. No specimen from Europe was found in the ‘Brazil’ clade. The geographic origin is indicated for each sequence; for France, the MNHN registration number of the specimen, the Department number and the administrative commune are indicated. The O. nungara clade includes many sequences that were deposited in GenBank under different names.

The O. nungara clade included 91 sequences and had high support (ML 96%, NJ 99%). Because of the complex nomenclatural history of the species, many specimens labelled in GenBank as ‘Obama marmorata’, and others labelled as ‘Obama sp.’ or ‘Obama sp. 6’, were included within the O. nungara clade—we consider these sequences as representatives of O. nungara.

Within the O. nungara clade, there were three clades, each with high support. These three clades are outlined in the simplified tree in Fig. 18. We chose the names for the clades according to the country of origin of sequences from South America.

Figure 18 Simplified tree of the relationships within members of Obama nungara and with close species.

This simplified tree is drawn from the complete tree shown in Fig. 17. There are three clades within O. nungara, representing three different populations within the species: ‘Argentina 1’, ‘Argentina 2’ and ‘Brazil’, each with high support. Invasive specimens found in most countries in Europe (Spain, Portugal, France, UK, Italy and Switzerland) were included in the ‘Argentina 1’ clade; however, a few specimens from Spain were included in the ‘Argentina 2’ clade. No specimens from the ‘Brazil’ clade were found in Europe.

The first clade (herein named clade ‘Argentina 1’, ML 89%, NJ 91%) is robust and includes most sequences of O. nungara, including specimens from Argentina and most specimens from European countries, including all specimens from France. This ‘Argentina 1’ clade is sister-group to a clade (with low support) which includes two clades with high support, a clade (herein named clade ‘Argentina 2’, ML 87%, NJ 86%) that includes specimens exclusively from Argentina and Spain, and a clade (herein named clade ‘Brazil’, ML 98%, NJ 98%) that includes exclusively specimens from Brazil.

Distances between and within the clades that constitute the O. nungara clade are detailed in Table 5. The mean distances within the clades were low, ranging from 0.39% to 1.18% according to the clade and method used, and the mean distances between clades were 3.62–5.39%, according to the clade and method used. Mean distances between the three O. nungara clades and the outgroup were higher, ranging from 9.25% to 17.61%, according to the clade and method used. This suggests that O. nungara is composed of three distinct clades, each with relatively low, albeit present, internal variation.

Table 5 Genetic distances between and inside clades of Obama nungara.

Results are given as percentages and mean distances, according to several methods (Tamura-3, Kimura-2 parameter, p-distance, Maximum composite likelihood). Differences between the various methods were minor, and the three clades of O. nungara were well separated between them and from the outgroup and had minor intragroup variation (in italics). The outgroup was composed of seven other species of Obama.

Method	Argentina 1	Argentina 2	Brazil	
Tamura-3	
Within group	0.40	1.15	0.53	
Argentina 2	3.89			
Brazil	4.32	4.78		
Outgroup	10.69	11.21	11.49	
p-distance	
Within group	0.39	1.12	0.52	
Argentina 2	3.62			
Brazil	4.07	4.47		
Outgroup	9.25	9.64	9.82	
Kimura-2	
Within group	0.40	1.15	0.53	
Argentina 2	3.84			
Brazil	4.31	4.76		
Outgroup	10.60	11.11	11.39	
Maximum composite likelihood	
Within group	0.40	1.18	0.54	
Argentina 2	4.11			
Brazil	4.89	5.39		
Outgroup	15.76	16.89	17.61	

Haplotype network

The matrix for the network included 91 sequences of O. nungara from nine countries; the matrix had 255 positions and was clean, without misread or blank parts. The haplotype network is shown in Fig. 19. The haplotype network analysis recognised 19 haplotypes for O. nungara. Three main networks were obtained.

Figure 19 Network of relationships between specimens of Obama nungara from various localities.

Three groups are visible: ‘Argentina 1’, ‘Argentina 2’ and ‘Brazil’. The main group, ‘Argentina 1’, includes sequences from Argentina and all invaded countries in Europe. The group ‘Argentina 2’ includes sequences from Argentina and a few from Spain, but no other country. The group ‘Brazil’ includes only specimens from Brazil, none from Europe. Within the group ‘Argentina 1’, three main haplotypes and seven minor haplotypes are visible; two of the main haplotypes include both specimens from Argentina and most invaded European countries.

The most important network in terms of numbers of sequences included 10 haplotypes, in which three were represented by more than 10 samples and the six others were either singletons 5 or with two samples 1. This large network included specimens from Argentina, and seven European countries including Spain, Portugal, France, Belgium, UK, Italy and Switzerland; it corresponded to clade ‘Argentina 1’ of the tree. The three main haplotypes within this network had different geographical compositions. The largest haplotype included representatives from Argentina and six European countries, the second had specimens from Argentina and four European countries, and the third had only specimens from Spain; among the small groups, five are from Spain and two from France.

Another network included five haplotypes, each with 1–2 specimens, and corresponded to clade ‘Argentina 2’ of the tree; specimens were only from Argentina (four haplotypes) and Spain.

One network, with four haplotypes, included only specimens from Brazil and coincided with the ‘Brazil’ clade found in the tree.

Finally, most specimens from Europe belonged to the ‘Argentina 1’ clade, with only a few specimens from Spain in the ‘Argentina 2’ clade; no specimens from Europe were found within the ‘Brazil’ clade.

Reports by citizen science and their significance

In Fig. 20, we report the number of findings of O. nungara in Metropolitan France each year from 2013 to 2018. Two peaks are visible, in 2014 and 2018.

Figure 20 Number of reports of Obama nungara from 2013 to 2018 in France.

The reports obtained from citizen science are indicated for each year; 2013 includes only half of the year. There was a total of 530 verified records. See text for comments about the probable biases which explain the higher numbers of records in 2014 and 2018.

In Fig. 21, we show the findings for O. nungara in Metropolitan France arranged by month, for all our records from 2013 to 2018. The histogram shows a tendency to a general peak in late spring (May and June) and another, weaker, peak in autumn (October); records during winter are minimal.

Figure 21 Monthly reports of Obama nungara from 2013 to 2018 in France.

All reports from 2013 to 2018 are here shown as monthly records. The highest number of records was obtained in May–June. See the “Discussion” for a possible bias for these records.

In our daily work at validating and compiling records from citizen science, our attention was drawn to the fact that mentions of land flatworms in the media were immediately followed by the arrival of numerous records, followed by a calmer period when media reports were not released. In Fig. 22, we show the variation of weekly records in 2018 for all species of land flatworms, including O. nungara (which represents about half of the records). The peak of weeks 21–24 in 2018 coincides with the publication of our article in PeerJ about bipaliines, which benefited from a press release and received strong interest from the media in France. These observations are commented in the “Discussion”.

Figure 22 Reports of alien land planarians in 2018 in France (all species including Obama nungara), shown as weekly numbers.

The total number of records in 2018 was 262. The peak in weeks 21–24 followed publication of our article in PeerJ, published 22 May 2018, and thus is not an indication of a higher number of animals in this period. See “Discussion” for temporal biases in citizen science.

Discussion

Populations of O. nungara and the invasion of Europe

Our results confirm the conclusions of Lago-Barcia et al. (2019) on the routes of invasion of O. nungara. In the continent of origin, South America, there are three known populations of O. nungara, two in Argentina, namely ‘Argentina 1’ and ‘Argentina 2’ and one in Brazil, namely ‘Brazil’. Only specimens with sequences identical or close to members of the populations ‘Argentina 1’ (in seven countries) and ‘Argentina 2’ (only in Spain) were found in Europe. The invasion of several countries of Europe has its origin in Argentina, not Brazil. Our analysis of available sequences, at the present time, suggests that the ‘Argentina 2’ population in Europe is still restricted to Spain, while the ‘Argentina 1’ population is widely present in most invaded European countries.

Our analysis confirms published findings (Lago-Barcia et al., 2019), with specimens from additional countries. Our matrix includes a higher number of sequences of O. nungara (91 vs. 66). We did not use exactly the same sequences and the same matrix, because of the limitations detailed in “Materials and Methods”, with sequences from different authors not overlapping; however, we found similar results for the origin of the invasion. Our matrix contains a lower number of sequences from Argentina, because we eliminated sequences with misreads, but this did not diminish the value of the interpretation. The networks in our analysis correspond well with those of the previous study, with ‘Argentina 1’ = ‘Network 3’, ‘Argentina 2’ = ‘Network 1’ and ‘Brazil’ = ‘Network 2’. In addition to the previous study, our study includes sequences from a larger number of countries and several new sequences from France. We found a total of 19 haplotypes (vs. 21) for O. nungara, including 10 haplotypes in the largest network, ‘Argentina 1’ compared to 11 in the corresponding ‘Network 3’ in the previous study. Our results therefore confirm and expand the conclusions of Lago-Barcia et al. (2019).

For future studies on O. nungara, COI sequences, with a large amount of data available, will certainly remain a primary choice, but more variable sequences will be useful for a finer analysis of the invasion.

The extent of the invasion in Metropolitan France and Europe

As shown on the maps (Figs. 14 and 15), O. nungara has invaded a large part of Metropolitan France and is found in 72 of the 96 Departments of Metropolitan France. The species was also recorded in mainland Europe in Spain, Portugal, Italy, Belgium, possibly the Netherlands, and Switzerland, and on European islands such as Guernsey, Great Britain, and Ireland in the North, and Corsica in the Mediterranean Sea (Table 2), as well as Madeira in the North Atlantic. The species is seemingly absent from Germany (Sluys, 2019), but given that several of our records in Metropolitan France are just a few kilometres from the German border, it is likely that the species will be recorded in this country very soon. No records were found from countries east of Germany.

Many reports of new invasions by various species of land flatworms are published every year (Chaisiri et al., 2019; Hu et al., 2019; Jones, 2019; Justine et al., 2018a, 2019; Prozorova & Ternovenko, 2018; Rodríguez-Cabrera & Torres, 2019; Soors et al., 2019; Vardinoyannis & Alexandrakis, 2019). As for other land planarians (Sluys, 2016), the origin of the invasion by O. nungara is likely to be the international trade of plants, since adult planarians and cocoons can easily travel in pot plants. After an invasion from Argentina (probably with multiple worms) to an unknown country (or perhaps several countries) in Europe, the species was then transmitted from country to country in Europe through intra-European plant trade, and again within each country by the same ways.

More than half of the records of O. nungara in Metropolitan France were from an altitude lower than 50 m, records above 250 m were very rare, and there was no record above 500 m (Fig. 16). While most of this altitude information might seem redundant with the geographical information (coastal localities are at low altitude), we believe that the absence of O. nungara above 500 m is significant. We also note that most, if not all, reports in other European countries are from localities at a low altitude; interestingly, the single report from Switzerland, a mountainous country, is from Geneva, with an altitude (369–458 m) below the limit found in France. Although more detailed studies are probably needed, we propose the hypothesis that the limiting factor is freezing, more likely at higher altitudes. This might be an important factor limiting the species in its invasion of Europe.

Our information on the local abundance of the species, although based only on data from non-professionals, is impressive. Hundreds of specimens in a single garden have been reported several times, and the counts by Michel Hir and Sylvain Petiet indicate that an invaded area can produce, by reproduction, an estimate of 1,000 adult individuals/day/hectare.

Geographical and temporal biases in citizen science data: a warning

Data from citizen science were instrumental in our research. It has been shown that recordings of invasive species is one of the major fields of research using citizen science (Kullenberg & Kasperowski, 2016). However, the quality of data obtained from citizen science requires assessment (Kosmala et al., 2016), and we would like to point out two biases in our data: geographical and temporal.

Geographical biases happen when more reports are obtained from an area where more members of the public provide reports, although this area does not actually harbour more specimens. This is evident on our map (Fig. 14) for the reports from the area around Paris. The Paris region harbours 18% of the French population, with a much higher density of people compared to the density in the rest of the country. However, it could be argued that cities are warmer than the countryside surrounding them, therefore providing better conditions for the species in winter. Two high-density regions on our maps (Fig. 15) probably reflect higher human population densities: the Gironde Department includes Bordeaux, the 6th largest French city in terms of population, and Haute–Garonne includes Toulouse Metro Area, the 4th largest urban area in France in terms of population. Another possible bias that is not easy to quantify, is the cultural difference between urban populations that are accustomed to using smartphones and social media, and rural populations, often older, where people are possibly less familiar with modern technology. We have anecdotal storeys about this cultural gap. In some rare cases, we received records of flatworms as photographs printed on article and sent by post—this was only from rural populations, not from cities. We also had discussions by telephone on how to ask a grandson to use his smartphone to record the flatworms found in his grandparents’ garden—the grandparents themselves having only a landline. Clearly, records from citizen science should be considered with this bias in mind. Cultural differences between urban and rural populations in France are probably relatively minor compared to what can be found in other countries; a recent study in Thailand found that more reports of P. manokwari were received from the great shore cities than from the inland (Chaisiri et al., 2019), which should be, in our opinion, interpreted more as a geographical/cultural bias, rather than a preferred location of the species.

Temporal biases occur when more reports are received at certain times of the year, independently of the actual abundance of the species. In our daily work in receiving emails from non-professionals, we noted that waves of records were received within days of certain media reports featuring our research (either radio, television or newspapers). The media effect was visible even though the species featured in the media was different (i.e. bipaliines, or P. manokwari, vs. other land flatworms). In fact, the emails we received often mentioned the recent media report which provided the information; this was even more visible when regional newspapers were involved, with waves of records coming from certain regions only. Figure 22 shows that weeks #22–24 in 2018 provided a high number of records; clearly, this was after our article about bipaliines was published (Justine et al., 2018b). This article was accompanied by a short article in French in the open-access media The Conversation (Justine, 2018), which was read more than 1,000,000 times in 1 week (Justine, 2019a), and produced a number of mentions in the national media (Bardou, 2018; Morin, 2018; Vidard, 2018) and the media in other countries (Gabbatiss, 2018; Guarino, 2018). However, May is also a period when O. nungara is abundant, and, in the Northern Hemisphere, a period when amateur gardeners spend much time in their garden. Figure 21 shows the number of records of the species each month during the 5 year survey. Although it is probably true that O. nungara is less abundant in winter (December–February) we do not know with certainty whether this is an actual decrease of abundance, or whether the number of records is low simply because this is a period when gardeners spend less time in their gardens. Similarly, the peak of abundance in May and June is probably genuine, but the data were biased by the abundance of records in May 2018 mentioned above (Fig. 22).

These geographical and temporal biases might hamper a thorough analysis of the data from citizen science; again, we insist that citizen science data are of extreme interest, but their interpretation should never neglect these possible biases.

Conclusion

In this article, we analysed data from citizen science over a five-and-a-half-year period and showed that O. nungara has invaded more than two-thirds of Metropolitan France, except for mountainous areas. We also found that local abundance in gardens can reach up to 100 of specimens, and that the specimens in an invaded garden can produce, by sexual reproduction, numerous progeny.

Our molecular barcoding study based on newly acquired and GenBank sequences of the COI gene confirmed that the populations of O. nungara found in Europe are similar to two populations from Argentina (but not from Brazil) (Lago-Barcia et al., 2019). We also confirmed or showed that a single population has invaded Spain, Portugal, France, UK, Italy, Belgium and Switzerland.

Although more than 10 species of land flatworm have invaded Europe (Table 1), no other species is comparable with O. nungara in terms of the size of the invaded territories: mainland Europe in Portugal, Spain, France, Italy, Belgium and Switzerland, islands in Europe such as Britain, Ireland and Guernsey, as well as Corsica in the Mediterranean, and Madeira in the North Atlantic; and in terms of recorded densities of animals in invaded areas. This clearly designates O. nungara as the most important invasive species of land flatworms in Europe.

Since the species has predatory habits on species of the soil fauna, its ecological impact is probably high, but has yet to be studied. Also, since the species seems to have a large spectrum of prey in its native area (Boll & Leal-Zanchet, 2016), including earthworms and molluscs, more studies are needed to determine which species, and in which quantities, are most preyed upon in Europe. Molecular methods (Cuevas-Caballé, Riutort & Álvarez-Presas, 2019) are available for obtaining this information.

Supplemental Information

Supplemental Information 1 All data obtained from Citizen Science (names of citizens have been removed).

Click here for additional data file.

Supplemental Information 2 Matrix used for the phylogenetic analysis, in two formats (FASTA and MEGA).

The matrix includes a selection of available sequences of O. nungara and sequences of six other Obama species (noted with *), selected upon the absence of misreads. The matrix includes 99 nucleotide sequences with 255 positions. The file in MEGA format (.meg) includes information about “groups” (a MEGA feature), that is the clades found in the analysis. The file in FASTA format (.fasta) does not include this information.

Click here for additional data file.

Supplemental Information 3 Matrix used for the network analysis.

This matrix includes only sequences attributed to O. nungara. It includes 91 taxa and is 255 bases in length. Traits are the geographical origin of the species, with number of traits (NTRAITS) set to nine, and trait labels set as: Argentina, Brazil, UK (including Guernsey), Portugal, Spain, Italy, Switzerland, Belgium, and Metropolitan France. NTRAITS are used by the PopART software (Leigh & Bryant, 2015).

Click here for additional data file.

Supplemental Information 4 French translation of the paper / Traduction intégrale en français.

Obama chez moi ! L’invasion de la France métropolitaine par le ver plat Obama nungara (Plathelminthes, Geoplanidae).

Click here for additional data file.

Supplemental Information 5 Infographic (in French).

Click here for additional data file.

Supplemental Information 6 Infographic (in French).

Click here for additional data file.

A complete French translation of the article is available as a Supplemental File / Obama chez moi ! L’invasion de la France métropolitaine par le ver plat Obama nungara (Plathelminthes, Geoplanidae). We thank all the members of the general public who participated in the survey; those who sent specimens are particularly thanked. Names of non-professionals, and sometimes scientists, who provided photographs and/or specimens are indicated in Tables and Figures. Two contributors, Sylvain Petiet and Michel Hir, kindly provided their results for counting. The support of various Regional federations for the control of pests (Fédérations Régionales de Défense contre les Organismes Nuisibles, FREDON) is acknowledged. Hugh Jones (NHM, London, UK) kindly provided unpublished information about records in the UK, Ireland and Madeira. Olivier Gargominy (MNHN, Paris, France) kindly identified a snail from photographs.

Additional Information and Declarations

Competing Interests

Author Contributions

DNA Deposition

Data Availability

Jean-Lou Justine is an Academic Editor for PeerJ.

Jean-Lou Justine conceived and designed the experiments, performed the experiments, analysed the data, prepared figures and/or tables, authored or reviewed drafts of the article, and approved the final draft.

Leigh Winsor conceived and designed the experiments, performed the experiments, analysed the data, authored or reviewed drafts of the article, and approved the final draft.

Delphine Gey conceived and designed the experiments, performed the experiments, analysed the data, authored or reviewed drafts of the article, and approved the final draft.

Pierre Gros conceived and designed the experiments, performed the experiments, analysed the data, prepared figures and/or tables, authored or reviewed drafts of the article, and approved the final draft.

Jessica Thévenot conceived and designed the experiments, analysed the data, prepared figures and/or tables, authored or reviewed drafts of the article, and approved the final draft.

The following information was supplied regarding the deposition of DNA sequences:

All new sequences are available in GenBank: MN529561–MN529582.

The following information was supplied regarding data availability:

The raw data from Citizen Science, the matrix used in the molecular phylogeny analysis and the matrix used in the molecular network analysis are available as Supplemental Files.

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
