# Peer review of "Obama chez moi! The invasion of metropolitan France by the land planarian Obama nungara (Platyhelminthes, Geoplanidae)"

_PeerJ, doi:10.7717/peerj.8385_

## Round 0.1 · original submission · Major Revisions

Apologies for a longer than expected review time, it was hard to find appropriate reviewers in summer. We have now received three detailed reviews and I suggest that you modify your manuscript based on their suggestions.

In addition, I would like to suggest that you involve a recognized expert in genetics, with a track record in sexual and asexual reproduction.

Please make sure that you give sufficient credit to relevant literature from other authors working on Obama nungara. Please make sure that you cite them appropriately in the introduction and discussion.

·

Basic reporting

Please see general comments to the author below.

Experimental design

Please see general comments to the author below.

Validity of the findings

Please see general comments to the author below.

Additional comments

Jean-Lou Justine et al 2019 PeerJ paper.

An interesting and valuable paper resulting from an extensive citizen science project. Publication is highly recommended, though some reorganisation and amendments are suggested.

I am not competent to comment in detail on the molecular part of this paper, though the conclusion seems reasonable.

Detailed comments follow.

Abstract
The Abstract could be much more concise without compromising the paper.
Is it usual to have sub-headings or even paragraphs in an Abstract?

Line 28. Delete “In this paper” and re-phrase this statement to something like:-
“A citizen science survey from 2013 to 2018 yielded 531 validated reports of the presence of O. nungara in 72 of the 96 departments of Metropolitan France.”
Rather than saying “over 500 reports” just give the number, 531.

Line 34: In English, Brittany is spelt with 2 ts.

Line 37: Pyrenees, not Pyreneans.

Line 45: Delete “The present findings strongly” Replace with “This suggests…”.

Throughout the MS: “O. nungara” after its initial mention unless at start of sentence.

Introduction
Much of the first paragraph is not needed.
Could start with something like:
“Obama nungara Carbayo, Álvarez-Presas, Jones & Rutort 2016 is one of several alien land planarian species found in Europe (Sluys, 2016).”

It is conventional to give the naming authority, Carbayo et al. 2016, after the first mention of the species. This should apply to all named species. Does this apply in PeerJ?

Lines 65-67. This is a conclusion, not an introduction!

Line 66: “land flatworm” not “land flatworms”.

Line 77: delete “the numerous”.

Line 81-2. Delete. This is a conclusion, not an introduction.

Line 90: Commas need inserting and removing to read as follows: “… generally by email and, since 2018, from a dedicated web page…”.

Line 105: “side” not “size”.

Line 109: “…is unlikely” not “are unlikely”.

Results
Some re-organisation is needed. Results from the citizen science data should be presented and analysed in full, followed by the results from molecular analysis (or vice versa). As it is, the the molecular results splits the former.
I question the need for the section on Morphology since the specimens are identified as the species Obama nungara.
Line 185: The “Tiger stripes” described are the result of dorsal pigment overlying each dorsal testis.
Line 195: The section on colour variation is good but it would be good to have some numbers. For example, the dark brown form is stated as most abundant – how many and what proportion of the total? Ditto for other colour varieties. Give data.

Line 199: Delete: “are medium sized” to read: Living adult specimens ranged in length..’

Line 207-210: This should be the first sentence in the Results section.

Line 316 and following. This section should precede the molecular section. There should be a section purely on the citizen science results, sub-divided as appropriate, followed by the molecular results.

Line 317-322: Re-organise this section to something like: “Over the the 5½ years of the project, more reports were received in 2014 and 2018 than in other years (Fig 19). On a monthly basis, more reports were received in May and June and also in October (Fig 20).

Line 323-330. Consider moving this to Discussion

Discussion
Line 329. “…commented upon…”.

Line 333-358. Should this come much later in the Discussion? Discuss the distribution first, then the molecular results – in the same order as the Results section.

Line 360: It is sufficient to state that “Obama nungara has been found in 72 of the 96 Departments of Metropolitan France (Figs 13, 14).
Delete “As shown on the maps” and “has invaded a large part of Metropolitan France and.”. These are redundant words.

Line 373. Records from above 250 m altitude were “very rare”. How many and what percentage of the total? Fig 15 might be more clear with a logarithmic Y-axis. Alternatively give the data as a table including percentages.
Abbreviate Figure 15 to Fig 15.

Line 378. Insert comma after bracket to read: “…(369-458 m), below…”.

Line 380: amend to “...freezing, more likely at higher altitude”.

Line 383: “…is impressive.”

Line 434: (Fig 21) not (Figure 21).

Conclusions
This is more a Summary than Conclusions. It is almost a repetition of the Abstract.
Line 439: Suggest: “Records over five and a half years resulting from citizen science show that O. nungara has been recorded from more than two-thirds of Metropolican France departements, with the exception of mountainous areas. Hundreds of of specimens may be found in a single garden. It is estimated that potentially about 1000 individuals per day per hectare could be produced by sexual reproduction.”

Line 440. “two-thirds” not “two third”.

Line 443. Delete “and” between “per day” and “per hectare”.

Figures.
Generally excellent.
Figure 20. Could error bars be included – or some indication of the difference between years? Perhaps the X axis could be months from start to finish over the 5½ years.

Table 2 Small amendment suggested as follows:
Ireland 2009 H. D. Jones, personal communication (col. & ID: R. Anderson).

Table 5. It is usual to give Mean ± standard deviation.

·

Basic reporting

First at all, I want to congratulate the authors for this important and relevant piece of work. The manuscript is original and provided many important information regarding the biology of an important alien species of European soils.
All the basic requirements are met. However, some important works are missing (e.g., Boll & Leal-Zanquet, 2016; Lago-Barcia et al., 2015) and many previously known facts are presented without referring to the appropriate previous work. Please, find more comments on this topic in the section “General comments”.

Experimental design

All the requirements are met, especially for the citizen science stuff of the manuscript. However, there are many errors and things which are unclear in the molecular part of the manuscript. It is unclear if the use a subset of the sequence of the species in Europe or all the available sequences (some key sequences are missing, see the “General comments section” for more information) and also the length of the sequences used in the analyses.

Validity of the findings

The results obtained in this publication will have an important impact on the study of exotic terrestrial flatworm biology, which represent an important concern for local soil fauna biodiversity conservation. Although the information from the citizen science project is in my opinion beautifully worked, the molecular part of the work is far more less novel and is flawed by some methodological and interpretation problems. Thus, the molecular part of the work critically needs a throughout revision before publication.

Additional comments

I really liked this manuscript. Please, let me congratulate you for the large amount of material you were able to collect and the impressive way you engaged citizen scientists to work with you. My recommendation is the publication of the manuscript after the authors perform a throughout revision. I will structure my review in two main parts: “major” and “minor” concerns. Major concerns are important errors that must be corrected before publication. Minor concerns are suggestions and minor comments that the authors should fix.
MAJOR CONCERNS
1. In the abstract the authors stated that they analyzed the COI variation of European specimens of the species collected from France, Italy and Switzerland. Thus, some parts of the distribution range of the species were not covered. However, in line 275, they said they used all available sequences on GenBank. Please, clarify which option was used. There are also published sequences of the species from Spain and Portugal (Álvarez-Presas et al., 2014; Lago-Barcia et al., 2015 and 2019). In my opinion, sequences from specimens from every part of the species distribution range (including native and invaded places) should be used. Please, add them and add the correct reference for each sequence.
2. Abstract and elsewhere: “This suggests that a single clade from South America invaded Europe, and that this clade had its origin in Argentina”. This statement is wrong and repeated many times in all the manuscript. As Lago-Barcia et al. (2019) demonstrated, the European specimens belong to both clades Argentina 1 and 2 (see major concern 3 for clarification regarding the nomenclature of the populations), and European populations were formed by at least 2 independent invasions of members each clade. Correct in every part of the manuscript.
3. The literature regarding the clades (=populations?) of Obama nungara is extremely confusing since at least three unrelated research groups worked independently on them and applied different nomenclatures without taking into account other previous work from other researchers. As an author of 2 of the articles dealing with this species in Europe, I am responsible of part of this mess, which I humbly accept. I think this interesting manuscript by Dr. Justine and coauthors brings a great opportunity to fix this problem in a definitive way. Here, the authors applied new names for the 3 previously known clades of Obama nungara (see Lago-Barcia et al., 2019 for more info): Obama nungara clades “Argentina 1”, “Argentina 2” and “Brazil”. Previously, the very same clades have been named as Obama nungara “network 3”, “network 1” and “network 2”, respectively. Álvarez-Presas et al. (2014) named “Argentine 1” and “Argentine 2” as “Obama sp. A” and “Obama sp. B”. Lago-Barcia et al. (2015) identified as Obama marmorata members of “Argentine 1”. Carbayo et al. (2013) named the members of the Brazilian clade as “Obama sp. 6” and later described Obama nungara using sequences from all the three lineages in Carbayo et al. (2016).
I understand the new nomenclature provided by the authors is far more attractive than “networks 1, 2 and 3”, so I encourage them to stick to this nomenclature, although it is new. However, I think they should properly refer to the previously literature identifying all these clades and also provide a synonym list for them to avoid more confusion in the future with this species. This is especially relevant for Table 1, where an incomplete list of synonyms is provided for Obama nungara and for understanding why so many sequences inside the Obama nungara clades of figure 16 are labelled as “Obama marmorata” and other names.
4. If the most fitting model was Tamura 3-parameter model with Gamma, why did the authors provide K2P genetic distances instead of uncorrected distances (p-distances)? Please, provide uncorrected distances for distance-based inferences. Also, use the most fitting model for the neighbor-joining analyses or p-distances, but not another model. How did you test if it is the proper model? State the software and methodology. Also use the same model for the ML reconstruction.
Please, be aware that in the results section (L. 268-272) the p-distances are provided, but in the material and methods is not stated that they were calculated.
I would say that the “evolutionary history” was assessed when using coalescent or other similar methods, which are more “historical” phylogenetic methods. Can you instead indicate “the phylogeny was inferred” instead.
5. Please, provide a full description of your haplotype networks. How many haplotypes did you find? Were they equally abundant? How many mutations were present between the networks and haplotypes inside the same network? Did you find the same number of haplotypes in the native and invaded regions? Did you find private haplotypes for any of those regions? Use all this information to discuss it in the context of the results obtained with the same methodology by Lago-Barcia et al. (2019).
6. “Interpretation of trees and network for the origin of invasion in Europe”. This section does not represent result but interpretation of them. It should be placed in the discussion. Besides, the results are biased since they did not use several Iberian sequences belonging to the Argentina 2 clade published by Álvarez-Presas et al. (2014) as Obama sp. B and later used by Lago-Barcia et al. (2019), grouping inside network 1 (see major concerns 2 and 3).
7. Authors should also properly acknowledge the previous literature. For instance, in L. 333-239 (as currently written), the authors attribute to themselves the same results obtained previously by Lago-Barcia et al. (2019). Besides, the last sentence is wrong, since Lago-Barcia et al. (2019) identified at least 2 different invasions events of Europe from two different populations coming from Argentina (networks 1 and 3).
8.I would not go so far on the arguments based on sexual and asexual reproduction used to explain the genetic variability of Obama nungara. Species with sexual reproduction might have extremely low levels of COI variation, even with a single haplotype described although many specimens sequenced (many examples currently described in the literature). The mutation rate could be not equivalent for the same marker in different species (or subfamilies!!!). Asexual species with large genetic distances is also possible, since the increase of multiple neutral genetic mutations are expected to occur in asexual organisms due to the absence of genetic recombination and the accumulation of mutations (see Ament-Velasquez et al., 2016).
If the mitochondrial hereditary is exclusively matrilineal (is it in plathyhelminthes? does exist studies on that?), the mitochondrial genome is not expected to recombine, so sex is not producing genetic variation for mitochondrial markers by itself. On the other hand, mitochondria usually have weak or inexistent mutation reparation mechanisms, so genetic variability might perfectly arise and transferred to clones in asexually-reproducing species.
The absence of genetic variation in some species of alien terrestrial flatworms in the invaded areas might be better explained by the number of invasions and the number of individuals involved in those invasions. For instance, if a single fertilized female of Platydemus manokwari, a sexually reproducing species according to the authors, was the origin of the invasions in the remaining world (e.g., a single individual taken from the native area which reproduced in a garden center and them spread to others by plant commerce with the mechanisms outlined in L. 369-371) and assuming matrilineal inheritance, the observed pattern might be explained. For answering this question, in my opinion, other markers with a faster evolutionary rate (microsatellites? SNPs?) should be used. On the contrary, invasions might also have occurred one or several times with a more diverse number of individuals, producing a different pattern of diversity for COI. We also need to take into consideration our knowledge on the genetic variation of these species in the native areas, which commonly is unknown or poorly known for alien terrestrial flatworms. Sadly, it is the case of Obama nungara.
I suggest authors delete the whole paragraph or provide a better discussion using multiple references on sexually- and asexually-reproducing groups of species. If the second is followed, please do not restrict the reference to Platyhelminthes.
Since I was not formally trained as a genetist, I would suggest the editor to ask other reviewers their opinion if this part of the discussion is maintained in subsequent review rounds.
9. I do not understand how it is possible that the support values of the clades from the ML analysis are so low in this manuscript when using the same methodology Lago-Barcia et al. (2019) obtained far better results. In the current state, the phylogenetic tree of figure 16 reads as a basal polytomy for all the clades involved and even the clades they describe are not supported. Please, review your analysis and update them. Also consider the missing sequences of Obama sp. B from Álvarez-Presas et al. (2014).
10. Information in table 1 regarding the synonyms of Obama nungara is incomplete. Specimens from the Argentina 1 clade were referred as Obama marmorata in Lago-Barcia et al (2015). Specimens of both Argentina 1 and Argentina 2 clades were cited as clades A and B by Álvarez-Presas et al. (2014). I think this article give the authors a great opportunity to clarify all this nomenclature.
Obama nungara clade Argentina 1. Synonyms. Obama sp. A sensu Álvarez-Presas et al. 2014. Obama marmorata sensu Lago-Barcia et al. 2015. Obama nungara network 3 sensu Lago-Barcia et al. 2019.
Obama nungara clade Argentina 2. Synonyms. Obama sp. B sensu Álvarez-Presas et al. 2014. Obama nungara network 1 sensu Lago-Barcia et al. 2019.
Please, also keep in mind that many species were transferred to the current genus by Carbayo et al. 2013 and they might been recorded with other name in earlier publications. Those names should be reported as synonyms in this table.
11. Figure 17. Argentina 2 clade also includes individuals from Spain (see Lago-Barcia et al. 2019 for more information).
12. Information from Table 2. Also include other records of the species, such as those from Lago-Barcia et al. 2015 and Ávarez-Presas et al. 2014. I might forget more works reporting specimens, so this should be complemented with a more exhaustive literature search.
13. Please, revise the results from Table 2. They seems to be a great overlap between minimum and maximum interclade distances and when I did similar calculations with the similar matrix from Lago-Barcia et al. (2019) (unpublished results) I obtained a very different result. Also provide the results in percentage, are easier to read.
MINOR CONCERNS
L. 25. “Obama nungara has invaded several countries of Europe (Spain, Portugal, France, Belgium, UK, Italy)”. Please add “and” between “UK” and “Italy”. Remove the comma.
L. 45. “The present findings strongly suggest that Obama nungara is a highly invasive species and that the population which has invaded several countries in Europe comes from Argentina.” This conclusion is not novel from this work, so the authors should change this phrase to something like “The present findings strongly supports” or add “as previous research have shown”. Keep in mind that there is no single invasion of Obama nungara in Europe: there are 2. See the major concern number 2 and 3 for more information.
L. 58. Coenoplana coerulea is not rare at all, at least in Spain (unpublished observation of my own, but see the large amount of individuals sequenced by Álvarez-Presas et al., 2014). Please, reconsider this statement.
L. 61-64. Please refer to the specific works reporting this species in each of the listed places. They will help the readers to locate specific literature. Please, keep in mind that for some countries there more than an article dealing with them and have them compiled here would be very much appreciated for some readers (for instance me!  ). If you think this would make this paragraph less attractive to read, you can also add a table for that.
L. 66. Is there any previous research on abundance of the species in Europe? If so, please refer it.
L. 71-74. I do not get what you want to explain in this phrase. Can you reword it in a more clear way? Also, the placement of the reference seems to me maybe a bit misleading.
L. 75-76. Substitute “the name Obama nungara itself was assigned to the species as recently as 2016 (Carbayo et al. 2016)” by “the species Obama nungara Carbayo et al. 2016 was recently described” or a similar sentence.
L. 81-82. Remove. This is a conclusion of the work, not part of the introduction. Besides, the statement regarding the origin of the European populations is wrong (see major concern 2).
L. 86. Same methodology?
L. 87. Twitter.
L. 117. photographs.
L. 118. I think researchers usually need to get used to time consuming processes! Although I acknowledge all the time you spend making sure the rights of the authors of the photographs were fully respected, I do not see the necessity to add this phrase.
L. 124. (Switzerland and Italy).
L. 130. they were EUTHANATISHED in hot water and preserved in 95% ethanol.
L. 136-138. Are you renaming the pair of primers (JB3 and JB4.5?). Please, stick to the original nomenclature.
L. 138. “(Bowles et al. 1995; Littlewood et al. 1997)”. If those references refer to the both pair of primers in a respective way, please add “respectively”.
L. 139. “3Mm MgCl2”. I do not understand this measure.
L. 250. Substitute “plot” by “chart”.
L. 252. Space between the number and the “m” or not? Homogenize.
L. 307. If you use the Spanish sequences from Álvarez-Presas et al. (2014) and Lago-Barcia et al. (2019), you will see this statement is wrong.
L. 348. Reference missing.
No specific lines. In general, the discussion is weakly supported by references to other previous works which find/suggested the same. For instance, the plant commerce is commonly suggested as an invasion mechanism for soil fauna. The text would be stronger if the use previous literature to support the authors views and not present them as original thinking from the authors, especially if they cite also other researchers.
I personally love the section “Geographical and temporal biases in citizen science data: a warning” from the discussion. If possible, please provide more references on biased data from similar studies involving citizen scientists.
L. 423. 1,000,000 times
L. 427-428. “period when amateur gardeners spend more time in their garden”. Was this inferred from your observations, from your communication with citizen scientists or it is a suggestion from the authors?
L. 446. Correct the error regarding the geographic origin of the both Argentinean clades found in Europe.
L. 447-449. This information is not exclusive from your study. Rewrite.
L. 452-453. What exactly means the most important species? Careful: a less abundant species with a more specialized diet might have more important impact on native soil fauna by triggering local extinction of some species.
L. 454-458. Boll and Leal-Zanchet (2016) studied the diet of several species of land flatworms in their native area. Obama nungara was also assessed, showing a generalist predatory behavior. Please, include this information here, evaluate the putative impacts this kind of diet might have on native soil fauna and cite the article.
Figure 2 caption. Please, also include here the name of the person who identified the snail.
Figure 13 caption. Please, substitute “Mountainous parts (yellow on the map) were not invaded” by “There are not known records in the mountainous parts (yellow on the map)”.
Figure 14 caption. Please, substitute “The color shows the intensity of the invasion” by “the intensity of the color shows the density of recorded individuals by citizen scientists”.
Figure 16 caption. “The percentage of trees in which the associated taxa clustered together is shown next to the branches”. Is this an weird way to explain what bootstrap is? I do not fully understand it, please rewrite.
114 sequences. Remove “nucleotide” from here since it confuse the reader.
“150 positions in the final dataset” Do you mean nucleotides? In the material and methods longer sequences are described! This might be the origin of the low support values found in this study respective to Lago-Barcia et al. (2019).
Figure 16. Redo the analysis including the Álvarez-Presas et al. 2014 sequences of the Obama sp. B. You will see that also Argentina 2 specimens are present in Europe.
Figure 17 and figure 17 caption. Same comment as for figure 16. Redo and correct.
Figure 18 and figure 18 caption. Same.
Table 5. What means 20 days? 20 days in may? 20 days of total sampling? Clarify. Also provide how many days each month was sampled.
All the text. Please, provide the correct reference for Lago-Barcia et al. (2019), currently cited as Lago-Barcia et al. (2018, the online first version of the article). You can find here the correct bibliographic details: https://link.springer.com/article/10.1007/s10530-018-1834-9
REFERENCES (only listed if not currently present on the manuscript):
Ament-Velasquez, SL; Figuet, EJ; Ballenghien, M; Zattara, EE; Norenburg, J; Fernández-Álvarez, FÁ; Bierne, J; Bierne, N & Galtier, N. (2016) Population genomics of sexual and asexual lineages in fissiparous ribbon worms (Lineus, Nemertea). Molecular Ecology. DOI: 10.1111/mec.13717.
Piter Kehoma Boll, Ana Maria Leal-Zanchet. (2016) Preference for different prey allows the coexistence of several land planarians in areas of the Atlantic Forest. Zoology http://dx.doi.org/10.1016/j.zool.2016.04.002
Carbayo, F., Álvarez-Presas, M., Olivares, C. T., Marques, F. P. L., Froehlich, E. M., and Riutort, M. (2013). Molecular phylogeny of Geoplaninae (Platyhelminthes) challenges current classification: proposal of taxonomic actions. Zoologica Scripta 42, 508–528. doi:10.1111/zsc.12019
Lago-Barcia D, Fernández-Álvarez FA, Negrete L, Brusa F, Damborenea C, Grande C, Noren˜a C (2015) Morphology and DNA barcodes reveal the presence of the non-native land planarian Obama marmorata (Platyhelminthes: Geoplanidae) in Europe. Invertebr Syst 29:12–22.

Reviewer 3 ·

Basic reporting

A very interesting, well-researched and well-documented and well-written study. All figures are necessary.

Experimental design

The use of citizen science data is very good and up-to-date. It is very commendable that the authors also performed a molecular study for tracing the place of origin of the invasive species.

Validity of the findings

Findings and conclusions are supported by the data.

Additional comments

On a print-out of the manuscript I have made hand-written annotations (see attachment). Most of these concern suggestions for typographic and linguistic improvements, as well as a few corrections of references. I presume that these hand-written comments will be legible as well as intelligable to the authors. Attaching a pdf with my annotations considerably shortens the review process (for which reviewers were only given 9 days!). Providing these annotations in digital format would have greatly increased the necessary time for reviewing this manuscript. I hope that the authors will forgive me and will appreciate my detailed reading of the manuscript and the suggestions for improvement.

Annotated reviews are not available for download in order to protect the identity of reviewers who chose to remain anonymous.

---

## Round 0.2 · Minor Revisions

The manuscript has been significantly improved. Both reviewers were very positive about the quality of your work and the efforts that you have invested in the improvement of your manuscript. There are a few remaining minor points, mostly fixing typos. Please address them at your earliest convenience and resubmit the manuscript.

·

Basic reporting

See below.

Experimental design

No comment.

Validity of the findings

See General comments below.

Additional comments

Justine et al. 2nd review. Hugh Jones.

Much improved! Should be published. A couple of minor points follow.

Methods
Delete ‘five and a half year”. This is duplication since it is stated that the survey was conducted from “mid-2013 to the end of 2018”. Suggest giving the months and year of the survey, eg. July 2013 to December 2018.
This has consequences for Figures 20 and 21 (suggest make this more clear in each Figure legend).
In Figure 20, the 2013 the data is from only half a year, for the other years it is from 12 months (this is commented on in the legend as “half of the year” – give months, presumably July to December).
In Figure 21, the data for January-June covers five years, that from July-December covers six years. Could be given as mean+-SD error bars for each month.

Results.
Page 224. The dark brown form is stated as the most abundant form, “most abundant” covers everything from 51% to 99%!!!!

Table 1 and References. “Jones 1999” should read “Jones HD & Gerard BM. 1999”

Caenoplana bicolor should be called Caenoplana variegata.

·

Basic reporting

Everything is correct. I would also like to point out to the high quality of Suppl. material 2. The idea of adding both files, both in FASTA and in MEGA format, the last one including the grouping of each sequence, is glorious. Supplemental file 3 is also very useful to researchers willing to further research in this topic.

Experimental design

Everything is correct.

Validity of the findings

Everything is correct.

Additional comments

First at all, I want to give a very positive feedback to the authors, since the manuscript (which was very interesting in the previous version) is now greatly improved. I also acknowledge the time and efforts that the authors invested in providing excellent specific answers for all the reviewers´ comments. My recommendation is the publication of this brilliant piece of work. Happily, I only have a few minor suggestions that might improve the quality of the manuscript. “L. X” refers to the lines on the PDF reviewing file.
L. 28. cytochrome c oxidase 1
L. 60. (European_Union 2019). What is this? A citation?
L. 224-227. Do authors think the light and dark brown collorations have anything to do with the size of the specimen (i.e., larger specimens get more pigmented and change the colour). This question is more about personal interest than a comment for the manuscript.
L. 228. Living adult specimens range in length from 52 mm to 108 mm.
L. 289-290. If the authors have this information, it would be interesting to state if the 2019 observations refer to the same period of the year than the 2014 (May to August).
L. 308. (ML, 96 %; NJ, 99 %). Also applies to L. 315, 319 and 320.
L. 327. Delete “independent”, it is redundant referring to “clades”. Other alternative is substitute “independent clades” by “independent groups”.
L. 336. You might mean “individuals” or “samples” here, instead of sequences.
L. 336. I am not sure if “large network” is a correct expression.
L. 339. What does “the largest haplotype” mean? Maybe “the most represented”?
L. 341. Among the less represented groups.
L. 353-356. Please, provide more data in this part regarding the number of specimens found.
L. 371-372. Please provide your hypothesis or a citation explaining why it is clarified that the invasion is not from Brazil.
L. 382. In my opinion, making this effort to clean up the lower quality sequences increases the value of your interpretation!
L. 383. “Argentina 2”. Please, clearly state from where the nomenclature outside this work came from. I totally understand the meaning, but other readers less familiarized with the Obama nungara literature might find themselves totally lost here. Please, also relate your new nomenclature to the one from Álvarez-Presas et al. (2014) and Carabayo et al. (2013) explained in my previous report.
L. 386. The most represented network?
L. 389. COI sequences with.
L. 424. an estimate of 1000 adult individuals/day/hectare in these gardens. [Maybe other kind of area or in other place might produce less or more individuals]
L. 435-436. However, it could be argued that cities are warmer than the countryside surrounding them, likely providing better conditions for the species in winter. [Do we know which are the best conditions for these animals to live?]
Fig. 17. I think the sequence KJ659670 came from UK, not Spain. Add percentages to the bootstrap support values.
Table 1 caption. Dolichoplana striata Moseley, 1877 and Kontikia andersoni Jones, 1981.
Table 1. von Graff 1899;Winsor, 1991 #4083}. Probably the reference manager created some problems here.
Table 2. Why for some countries is only stated the first occurrence and in others also the work of first occurrence of the species and “this work”? Is the year of these two records the same?
Table 2. It might be useful isolate the record of Corsica from mainland France, since the first one represents an Island without a geographic continuity with mainland France and maybe this date might provide some insights (or not!) about the time of spreading of this particular invasive species.
Table 3. The specimen JL067 was received as a cocoon(s?). Do these sequence came from a single specimen isolated from the cocoon content?
Table 3. What does the “Code” column mean? Is it a political definition of the territories? If so, explaining it might help non-French readers to understand it better.
Table 5. p-distance and Maximum Composite Likelihood)

---

## Round 0.3 · accepted · Accept

It appears that the authors have addressed, or at least, considered, every comment raised by demanding reviewers. I recommend that this work is accepted for publication.